# Mixed-Strain Fermentation Conditions Screening of Polypeptides from Rapeseed Meal and the Microbial Diversity Analysis by High-Throughput Sequencing

**DOI:** 10.3390/foods11203285

**Published:** 2022-10-20

**Authors:** Wei Huang, Haining Xu, Jiayin Pan, Chunhua Dai, Benjamin Kumah Mintah, Mokhtar Dabbour, Rong Zhou, Ronghai He, Haile Ma

**Affiliations:** 1School of Food and Biological Engineering, Jiangsu University, 301 Xuefu Road, Zhenjiang 212013, China; 2Institute of Food Physical Processing, Jiangsu University, 301 Xuefu Road, Zhenjiang 212013, China; 3CSIR—Food Research Institute, Accra P.O. Box M20, Ghana; 4Department of Agricultural and Biosystems Engineering, Faculty of Agriculture, Benha University, Moshtohor P.O. Box 13736, Egypt; 5Fishery Machinery and Instrument Research Institute, Chinese Academy of Fishery Sciences, 63 Chifeng Road, Yangpu District, Shanghai 200092, China

**Keywords:** reseed meal, med-strain fermentation, polypeptide, microbial diversity

## Abstract

Conventional fermentation of rapeseed meal has disadvantages such as sterilization requirement, high energy consumption and low efficiency, as well as poor action of single bacteria. To overcome these drawbacks, mixed-strain fermentation of unsterilized rapeseed meal was investigated. Mixed-fermentation of unsterilized rapeseed meal (ratio of solid–liquid 1:1.2 g/mL) using *Bacillus subtilis*, *Pediococcus acidilactici* and *Candida tropicalis* (at 40 °C, for 3 days, with inoculation amount of 15% (*w*/*w*)) substantially increased the polypeptide content in rapeseed meal by 814.5% and decreased the glucosinolate content by 46.20%. The relationship between microbial diversity and physicochemical indicators showed that the improvement in polypeptide content was mainly caused by *C. tropicalis* (on the first day of fermentation) and *B. subtilis* (on the second day). Compared to raw rapeseed meal, the microbial diversity following the fermentation was significantly reduced, indicating that mixed-strain fermentation can inhibit the growth of miscellaneous bacteria. The study findings suggest that mixed-strain fermentation could be used to considerably increase the polypeptide content of unsterilized rapeseed meal, increasing the potential of rapeseed meal.

## 1. Introduction

Oilseed rape, also known as colza (*Brassica napus*), is a cruciferous annual herb whose seeds are rich in oil. Therefore, it is considered one of the top ten oil crops in the world. Among the world’s leading rapeseed producers, China ranks third with 14.05 million tons in 2020/2021 [1]. *Brassica napus* is pressed to extract rapeseed oil, and the remaining part is called rapeseed meal [2]. Rapeseed meal is rich in crude protein (with a balanced amino acid composition) [3], vitamins and minerals. Apart from it being treated as waste, rapeseed meal is mostly used as animal feed [4,5]. That notwithstanding, the proteins in rapeseed meal are macromolecular, which are difficult to digest and absorbed by animals. The rapeseed meal also contains anti-nutritional chemicals such as glucosinolate and phytic acid, which can damage the organs of animals [6,7]. In order to effectively utilize rapeseed meal, physical [8], chemical [9], enzymatic hydrolysis [10] and microbial methods are usually used to process rapeseed meal. Among them, the microbial method is simple to use, economical, and environment friendly; and has attracted the attention of researchers lately. Microbial fermentation uses microorganisms to ferment raw materials, which improves the quality of the raw material through a series of physicochemical reactions. There are two types of microbial fermentation methods, single-strain fermentation and mixed-strain fermentation. Single-strain fermentation typically uses mold or bacteria for fermentation. Shi et al. [11] observed that rapeseed meal fermented by *Aspergillus niger* had a notably high content of crude protein, small polypeptides and amino acids, and low content of various anti-nutritional factors compared with unfermented rapeseed meal. Liu et al. [12] fermented rapeseed meal with *Campylobacter sp. NJNPD41* and noticed that total free amino acids and short peptides yield increased by 115%, whereas the content of isothiocyanates and oxazolidinethione decreased by 53% and 60%, respectively.

Although single-strain fermentation can degrade some anti-nutritional factors, the structure of anti-nutritional factors in rapeseed meal is complex and diverse [13]. Single-strain fermentation alone cannot achieve the objective of complete detoxification, which requires the combined action of multiple microorganisms. Mixed-strain fermentation is the use of enzymes and secondary metabolites secreted by a variety of microorganisms for fermentation, which can better degrade anti-nutritional components of raw materials and consequently increase the nutritional value [14] and can inhibit the growth of miscellaneous bacteria in unsterilized raw materials. Maehara Larissa et al. [15] fermented unsterilized bagasse using a solid-state technique and found that relative to single-strain fermentation, the co-fermentation of bagasse (using *Trichoderma reesei* and *Aspergillus oryzae*) increased the amount of glucose released by about 50%. Leidy [16] and Chen [17] used mixed-strain to ferment unsterilized peanut silage and oat round bale silage, respectively; it was found that the silage quality was significantly improved after fermentation, along with modifications in chemical composition. Yusuf [18] and Bao [19] also indicated that mixed-strain fermentation could considerably reduce the anti-nutritional factors and improve the nutritional value of fermentation broth over single-strain fermentation. Compared to traditional single-strain fermentation, mixed-strain fermentation eliminates the need to sterilize raw materials and reduces energy consumption for production, but its strains and fermentation conditions need to be further explored.

The species, quantity and interactions of different microorganisms play an important role in the quality of fermentation products [20]. Studies have shown that *Bacillus subtilis* can produce abundant proteases and antimicrobial peptides that kill pathogenic bacteria, which is beneficial to the degradation of macromolecular proteins in raw materials and improves the disease resistance of animals [21]. *Candida* can secrete a variety of enzymes, which can greatly increase the protein content in feed [22], promoting the growth of livestock and poultry and improving the ability to resist diseases. *P. acidilactici* can produce lactic acid, which can reduce intestinal pH and improve the palatability of the feed [23]. However, the functionality of microorganisms during fermentation is not well understood, so understanding the changes in microbial flora during fermentation has important implications for the fermentation industry.

In recent years, high-throughput sequencing technology has been widely used in food fermentation. Several studies analyzed the dynamic changes of microbiota in the process of food fermentation, such as pepper [24], soybean meal [25], and rice wine [26]. Yu et al. [27] used high-throughput sequencing to reveal different core functional bacteria in three successive fermentation stages of broad bean paste with chilli. Wang et al. [28] studied the changes of microbial flora in Zhenjiang balsamic vinegar by high-throughput sequencing and found that the bacterial flora richness and diversity changed continuously with the acetic acid fermentation and divided the acetic acid fermentation process into three stages according to the changing pattern of bacterial community structure in different fermentation stages. High-throughput sequencing technology is considered a reliable method of analysis. However, studies on changes in microbial diversity during rapeseed meal fermentation are limited.

Therefore, the purpose of this work was to prepare polypeptides by fermenting rapeseed meal with *B. subtilis*, *P. acidilactici,* and *C. tropicalis* under non-sterile conditions and use high-throughput sequencing technology to analyze bacterial and fungal microbial diversity during the fermentation process. The relationship between polypeptide content, physicochemical parameters and microbial diversity during fermentation was also studied to provide a certain theoretical basis and technological knowledge for subsequent rapeseed meal fermentation.

## 2. Materials and Methods

### 2.1. Solid-State Fermentation of Rapeseed Meal

The rapeseed meal (which was purchased from Jiangsu Zhongyi Feed Co., Ltd., Yancheng China) was crushed and passed through a 10-mesh sieve for use. *B. subtilis* (CICC 10089), *P. acidilactici* (CICC 23484) and *C. tropicalis* (CICC 1254), purchased from the China Center of Industrial Culture Collection (CICC), incubated for 2 days under optimal conditions, and mixed in a 1:1:1 ratio to form a three-strain activation solution [29]. 200 mL of sterile distilled water was added to 200 g of rapeseed meal and then mixed with different inoculation amounts of the three-strain activation solution, stirred for 2 min and then kept in 14×20 cm PE fermentation bags for subsequent anaerobic fermentation at 30 °C.

### 2.2. Single Factor Experiment for Mixed-Bacteria Fermentation of Rapeseed Meal

The effects of different fermentation time (1, 2, 3, 4, 5 d), temperature (20, 25, 30, 35, 40, 45, 50 °C), amount of inoculation (5, 10, 15, 20, 25% (*w*/*w*)), and solid–liquid ratio (1:0.6, 1:0.8, 1:1, 1:1.2 and 1:1.4 g/mL) on the preparation of rapeseed meal polypeptide were investigated. Following the fermentation process, the samples were dried at 60 °C, pulverized and then sieved (60-mesh) to obtain the powder. 0.5 g of rapeseed meal was added to 50 mL of 15% trichloroacetic acid solution, stirred for 30 min, and thereafter centrifuged at 2000 rpm for 10 min. The polypeptide content in the supernatant of the various samples was determined by the Kjeldahl method [30].

### 2.3. Optimization of Fermentation Conditions

After the single-factor experiment, the best fermentation time, temperature, solid–liquid ratio and inoculation amount were selected and used to determine the optimal fermentation conditions of rapeseed meal based on the polypeptide content (experimental response). The central composite design (CCD) with the aid of Design-Expert 13 (Stat-Ease Inc., Minneapolis, MN, USA) was used to design a response surface analysis experiment with a four-factor-three-level, as listed in Table 1. After optimizing the fermentation conditions, a verification experiment was done, and the physicochemical parameters, as well as microbial diversity, were examined under the optimal fermentation conditions.

### 2.4. Physicochemical Parameters

#### 2.4.1. Determination of pH and Titratable Acidity Content in the Fermentation Broth

Five grams of the sample (taken every 24 h) was added to 10 mL of distilled water and afterward homogenized for 10 min. The pH of the sample was measured and then titrated with 0.1 mol/L NaOH for titratable acidity [31]; the endpoint pH was 8.2.

#### 2.4.2. Determination of Glucosinolate Content

The method of Liu et al. [32] with slight modification, was used to quantify the glucosinolate content of the fermented rapeseed meal. The content of glucosinolate was determined by the palladium chloride colorimetric method.

### 2.5. Determination of Microbial Diversity in Fermented Rapeseed Meal by High-Throughput Sequencing Technique

#### 2.5.1. Sample Preparation

Mark untreated rapeseed meal as RM. The rapeseed meal was mixed with distilled water and recorded as NI. The rapeseed meal was fermented with the strains under optimum conditions, and samples were taken every 24 h and recorded as C1, C2 and C3. Five groups of samples were snap-frozen in liquid nitrogen for 30 min and then stored at −80 °C. The microbial diversity of bacteria and fungi in the samples was determined separately.

#### 2.5.2. Microbial Diversity Analysis

DNA was extracted from fermented rapeseed meal using TGuide S96 Magnetic Soil/Stool DNA Kit (Tiangen Biotech (Beijing) Co., Ltd., Beijing, China), and then the concentration of nucleic acid was detected using a microplate reader (GeneCompang Limited, synergy HTX). After amplification, the PCR products were subjected to electrophoresis using 1.8% agarose to detect the integrity; and NanoDrop 2000 (Thermo Scientific, Waltham, MA, USA) was used to determine the concentration and purity of DNA.

Using the extracted DNA as a template, the bacterial 16S V3-V4 region was amplified using bacterial primers 338F (5′-ACTCCTACGGGAGGCAGCA-3′) and 806R (5′-GGACTACHVGGGTWTCTAAT-3′); and fungal primers ITS1F (5′-CTTGGTCATTTAGAGGAAGTAA-3) and ITS2 (5′-GCTGCGTTCTTCATCGATGC-3′) were used to amplify the ITS region of fungi. Construction of sequencing libraries and paired-end sequencing was performed on an Illumina NovaSeq6000 platform at Biomarker Technologies Co, Ltd. (Beijing, China) according to standard protocols.

To describe the microbial diversity, the Alpha diversity of bacteria and fungi was measured, and the specie abundance and diversity during fermentation were studied. Determining the role of microorganisms in different fermentation stages was done by studying the microbial diversity of bacteria and fungi at the phylum and genus levels.

### 2.6. Statistical Analysis

All experiments were repeated 3 times unless otherwise stated, and the results were displayed as mean (±) standard deviations. Origin 2021 (Originlab Corporation, North- 177 ampton, MA, USA) and SPSS 26 (SPSS Inc. Chicago, IL, USA) were used for data plotting and one-way ANOVA (*p* < 0.05), respectively.

## 3. Results

### 3.1. Effect of Single Factors on the Polypeptide Yield of Rapeseed Meal

#### 3.1.1. Fermentation Time

The effect of fermentation time on the content of rapeseed meal polypeptide is shown in Figure 1a. It can be seen that as fermentation time increased, the polypeptide content increased and then decreased, reaching a maximum on the third day of fermentation. This phenomenon may be related to the abundance of nutrients in the early fermentation medium. During this period the microorganisms grow vigorously, and a large number of proteases are produced to break down the proteins [33], causing an increase in polypeptide content. As the fermentation time increases, the nutrients in the medium become insufficient. Microorganisms use the available peptides or small molecule proteins to maintain growth and development [34], causing a gradual decrease in polypeptide concentration.

#### 3.1.2. Fermentation Temperature

From Figure 1b, it can be seen that the maximum polypeptide content in fermented rapeseed meal was obtained at a fermentation temperature of 40 °C. Too high or too low a fermentation temperature could inhibit the growth of microorganisms and thus impact the fermentation efficiency [35].

#### 3.1.3. Inoculation Amount

The influence of inoculation amount (5–25%) is exhibited in Figure 1c. It was observed that by increasing the inoculation volume, the polypeptide content increased first and then decreased, and the maximal value was noticed at 15% inoculation amount. This observation was possibly linked to the inoculation amount—when it was small, the number of microorganisms in the medium, as well as the protease produced was also small. A large amount of rapeseed protein is gradually broken down into smaller polypeptides. Further, when the inoculum was >15%, a decrease in polypeptide content was observed. This phenomenon may be due to the fact that excess microorganisms consume nutrients rapidly, leading to a decrease in the number of polypeptides already produced. This is similar to the results of Hou’s [36] study on soybean meal.

#### 3.1.4. Solid–Liquid Ratio

As shown (Figure 1d), the peptide content reached a maximum at a solid–liquid ratio of 1:1.2 g/mL and gradually decreased beyond 1:1.2 g/mL. At a relatively low solid–liquid ratio (<1:1.2 g/mL), insufficient water affects the exchange of substances between microorganisms and slows down metabolism. Too high a solid–liquid ratio (>1:1.2 g/mL) reduces the gas between substrates, thus limiting bacterial growth and enzyme production. Different solid–liquid ratios can affect the content of peptides [37].

### 3.2. Optimization of Polypeptide Yield from Fermented Rapeseed Meal

#### 3.2.1. Model Fitting

The process conditions of rapeseed meal fermentation were optimized by response surface analysis method. The central composite design and experimental results were shown in Table 2.

The quadratic regression analysis of variance was performed on the above experimental results, and the regression equation obtained is as follows:*Y* = *7.38* + *0.054A* − *0.09B* + *0.082C* − *0.21D* + *0.00625AB* + *0.016AC* − *0.025AD* + *0.049BC* − *0.069BD* + *0.027CD* − *0.43A*^2^ − *0.72B*^2^ − *0.73C*^2^ − *1.06D*^2^(1)

As can be seen from Table 3, the *R^2^* and *Adj-R^2^* of the model were 0.9926 and 0.9851, respectively, indicating the model has a high correlation between the measured and predicted values. Furthermore, the low coefficient of variation value (1.26%) suggested that the regression equation can be used to analyze and predict the experiment.

#### 3.2.2. Optimization of Fermentation Conditions and Model Verification

The optimum fermentation conditions were established to obtain the highest polypeptide yield. Accordingly, the optimal independent/ experimental variables (A = 40.33 °C, B = 1:1.189 g/mL, C = 15.26% and D = 2.902 d) were predicted to realize expected polypeptide content = 7.432%.

Considering the actual production operation, the optimal process conditions were modified to A = 40 °C, B = 1:1.2 g/mL, C = 15% and D = 3 d. Using these conditions, the measured mass fraction of peptides was 7.325%, which was close to the predicted value (7.432%), indicating that the constructed model made a good prediction for the preparation of peptides by fermentation of rapeseed meal.

### 3.3. Physicochemical Parameters

The pH of the fermentation substrate slowly decreased, while the titratable acidity content significantly increased with fermentation time, which may be related to the physiological function of the lactic acid bacteria (Table 4). At the beginning of fermentation, the number of *P. acidilactici* was not very high, inducing little changes in the pH and titratable acidity content of the fermentation substrate. By increasing fermentation time, *P. acidilactici* gradually reached a stable stage of growth, and the acid production reached a maximum, resulting in substantial changes in the pH and titratable acidity. On the other hand, the glucosinolate content of the fermented rapeseed meal was observably decreased by 46.20% at the end of the fermentation process compared to the unfermented sample (0 h). This degradation of glucosinolate may be credited to the synergistic effect of *P. acidilactici* and *B. subtilis* [38].

### 3.4. Changes in Bacterial Diversity during Fermentation of Rapeseed Meal

A total of 400,243 pairs of reads were obtained from the sequencing of the 5 samples. A total of 399,494 Clean Reads were generated after paired-end reads quality control and splicing. Each sample generated at least 79,505 Clean Reads, with an average of 79,899 Clean Reads.

The coverage rates of the five groups of all samples had more than 99.9%, indicating that the sequencing results can reflect the real situation of the bacteria in the samples. Chao1 and Ace indices measure species abundance, that is, the number of species. Simpson and Shannon parameters are used to measure species diversity. The larger the index, the higher the species diversity of the sample [39]. It can be seen from Table 5 that the alpha diversity index of the five groups of samples is different, and the highest value was noted for the RM group, indicating that the two different treatment methods can inhibit the growth of some microorganisms. Additionally, as the fermentation time increased, the indices of C1, C2 and C3 increased and then decreased, indicating that the microbial diversity increased and then decreased. This outcome may be attributed to the way microorganisms adapt to the external environment at the early stage of fermentation. At the middle stage of fermentation, their activity begins to stabilize, and thus microbial diversity increases. At the late stage of fermentation, there are fewer nutrients in the substrate, causing microbial growth to be restricted and diversity to decrease, which is also in line with the principle of pH and acid change.

The changes in bacterial diversity at the phylum and genus levels during rapeseed meal fermentation are shown in Figure 2a,b.

It can be seen from Figure 2a that, the dominant phyla in the RM group at the phylum level were *Firmicutes* (32.41%), *Proteobacteria* (23.46%), *Actinobacteria* (16.51%), and *Bacteroidetes* (8.23%), *Cyanobacteria* (4.92%), and *Chloroflexi* (3.59%). The dominant phyla in NI and C1, C2, and C3 groups were significantly different from those in the RM group. *Firmicutes* had always been the main dominant bacterial phylum in the C and NI groups.

Some studies have reported that *Firmicutes* is a common fermentative phylum [40,41,42], with the progress of fermentation, *Firmicutes* always occupy an absolutely dominant position and the proportion gradually increases. To further study the dynamic changes in the bacterial community during fermentation, the dominant bacterial community was further observed at the genus level. From Figure 2b, it can be seen that the dominant flora of each group of samples at the genus level has notable differences. Compared with RM, there were fewer miscellaneous bacteria in group C, indicating that the addition of bacteria inhibited the growth of miscellaneous bacteria. In the C1 (24 h), the proportion of the dominant bacteria (*Pediococcus*) was 56.99%, whereas the proportion of *Staphylococcus* and *Bacillus* was 24.98% and 5.52%, respectively. On the other hand, in the C2 (48 h), the proportion of the dominant bacterium *Pediococcus* was 34.29%, and the proportion of *Staphylococcus* and *Bacillus* was 30.07% and 25.18%, respectively. In the C3 (72 h), the proportions of *Pediococcus*, *Bacillus* and *Staphylococcus* were 47.88%, 29.99% and 17.09%, respectively. The percentage of other genera decreased from 12.51% to 5.04%. Both *Bacillus* and *Staphylococcus* belong to the *Bacillus* order of *Firmicutes*, and the appearance of *Staphylococcus* may be caused by the addition of *B. subtilis*. With prolonged fermentation time, the proportion of *Bacillus* initially increased and then decreased, and the abundance of *Pediococcus* decreased, first, and then increased. On the third day of fermentation, the proportion of the two bacteria groups was roughly the same, and the proportion of other bacteria had reached a very low level. Therefore, it can be seen from the flora ratio that on the first day of fermentation, *P. acidilactici* played a major role, and on the second day, it was *B. subtilis*, whilst *P. acidilactici* and *Bacillus*, together, played a major role on the third day. Studies have shown that *P. acidilactici* is a facultative-anaerobe [43]. The reduction of oxygen in the substrate is more suitable for the growth of lactic acid bacteria, increasing the abundance of *P. acidilactici*. The increase in lactic acid bacteria changes the pH and acid in the substrate, which can inhibit the growth of miscellaneous bacteria [44].

### 3.5. Changes in Fungal Diversity during Rapeseed Meal Fermentation

A total of 400,227 pairs of reads were obtained from the sequencing of the 5 samples. A total of 398,844 Clean Reads were generated after paired-end reads quality control and splicing. Each sample generated at least 79,624 Clean Reads, with an average of 79,769 Clean Reads.

It was observed from Table 6 that the coverage rates of the five groups of samples were all above 99.9%, indicating the sequencing results could reflect the real situation of fungi in the samples.

The changes in fungi diversity at the phylum and genus levels during rapeseed meal fermentation are shown in Figure 3a,b.

As shown in Figure 3a, the dominant phylum in each sample is *Ascomycota*, which was also found in the study of Wu [45], but the difference is that there are many unclassified fungal phyla in RM, while *Basidiomycota*, *Mortierellomycota*, etc. were identified in the other groups, which are similar to the results of Wei et al., [46]. Compared with group C, with the progress of fermentation, the proportion of *Ascomycota* remains stable and occupies a dominant position. The samples were further analyzed for changes in fungal diversity at the genus level, and the results are shown in Figure 3b. The dominant flora in RM is *Aspergillus* (32.13%) and some *unclassified Fungi*, while the dominant flora in each of the remaining samples was *Candida*, which belongs to the *Ascomycetes*. *Mortierella* (2.26%) and *Fusarium* (1.25%) were observed, indicating that they are more common in natural fermentation [47]. During the fermentation process, the proportions of *Mortierella*, *Fusarium*, and *Aspergillus* continuously decreased, which suggests *Candida* may have an inhibitory effect on other fungi [48]. The proportion of *Candida* increased and then slightly decreased, which may be ascribed to the reduction of nutrients in the matrix and the change in pH.

## 4. Discussion

From the changes in physicochemical parameters and bacterial flora during the fermentation process, it was concluded that *P. acidilactici* was the dominant bacteria on the first day of fermentation, but the pH and titratable acidity in rapeseed meal did not increase significantly. On the second day, the proportion of *P. acidilactici* in the flora decreased, and significant changes in the pH and titratable acidity were noticed. The reason for this phenomenon is that *P. acidilactici* may not have produced a lot of acids, during cell proliferation on the first day of fermentation. Also, the lactic acid bacteria reached a stable growth, and the *P. acidilactici* began to produce a large amount of acid at this time, resulting in significant changes in pH and acid. This phenomenon has also been reported by other researchers. Xiang [49] found that different lactic acid bacteria produce acid faster in the early stage of fermentation and some in the later stage of fermentation, and different lactic acid bacteria have different acid production periods. Similarly, Scholoss et al. [50] noted that *P. acidilactici* played a role in the 24–60 h acid phase of composting.

By linking the peptides prepared from rapeseed meal and the microbial diversity, it can be seen that the increase rate of peptides on the first day was the highest (212.4%). However, the increased rate of polypeptides on the second (68.19%) and third day (21.79%) gradually decreased. Previous reports showed that *B. subtilis* could produce abundant proteases, which can decompose macromolecular proteins into small molecules of a polypeptide or amino acids. But our observation (Figure 2b) indicated that *B. subtilis* does not occupy a large proportion on the first day of fermentation. Therefore, there may be other reasons for the increased polypeptide yield at this stage. Furthermore, *Candida* had an absolute advantage on the first day of fermentation (Figure 3b). As indicated, *Candida* also can produce protease, causing the increase in polypeptide yield on the first day. While *Candida* secretes proteases, it can also increase the production of various proteases together with *B. subtilis* [51]. On the second and third days of fermentation, although the increase rate of peptides decreased, the content of peptides continued to increase. Figure 3b exhibited that the proportion of *Candida* did not change significantly. Therefore, the increase in the content of polypeptides in the middle and late fermentation stages may be due to the gradual increase in the number of *B. subtilis* and the strong secretion of protease, increasing the content of polypeptide. This is also evidenced by the increased proportion of *subtilis* spores as shown in Figure 2b.

Glucosinolates have toxic effects on animals, and earlier studies indicated that lactic acid bacteria can degrade glucosinolates [52,53]. The degradation rate of glucosinolates was 11.16 and 28.40% on the first and second day of fermentation, respectively. Moreover, the proportion of *P. acidilactici* decreased in the first two days of fermentation, the proportion of *C. tropicalis* remained unchanged in the first two days, and the proportion of *Bacillus* gradually increased (Figure 2b and Figure 3b). Thus, the improvement of glucosinolate degradation rate on the second day may be due to the synergistic effect of lactic acid bacteria and *B. subtilis*, which was also reported by Zhang et al. [38].

On the last day of fermentation, the proportion of *Pediococcus* and *Bacillus* reached 77.87% (Figure 2b), and the proportion of *Candida* was 81.59% (Figure 3b). There were no microorganisms in absolute dominance in the control group. The microbial diversity of the control group was significantly higher than that of the test group, indicating that the fermentative bacteria had an inhibitory effect on the growth of other microorganisms [54].

## 5. Conclusions

In this study, we used mixed-strain microorganisms (*B. subtilis*, *P. acidilactici* and *C. tropicalis*) in fermenting unsterilized rapeseed meal to produce polypeptides to potentially augment the use of rapeseed meal in industry. The results showed mixed fermentation significantly enhanced the polypeptide content of rapeseed meal. Central Composite Design (CCD) was successfully employed to access the optimal conditions (fermentation temperature, time, inoculation amount, and solid–liquid ratio) for the preparation of polypeptide from fermented rapeseed meal. ANOVA illustrated that the experimental factors considerably impacted polypeptide yield. The maximal yield of polypeptide observed at the optimal fermentation conditions was close to the predicted value, implying the optimized fermentation process was feasible. High-throughput sequencing analysis indicated that mixed-strain fermentation could inhibit the growth of miscellaneous bacteria in rapeseed meal. This study can help in controlling the fermentation process and provide a basis for the further utilization of rapeseed meal. This article is based on mixed-strain. However, the theoretical mechanisms of association between individual strains and metabolites, as well as the coordination of flavor by mixed strains should still be studied.

## Figures and Tables

**Figure 1 foods-11-03285-f001:**
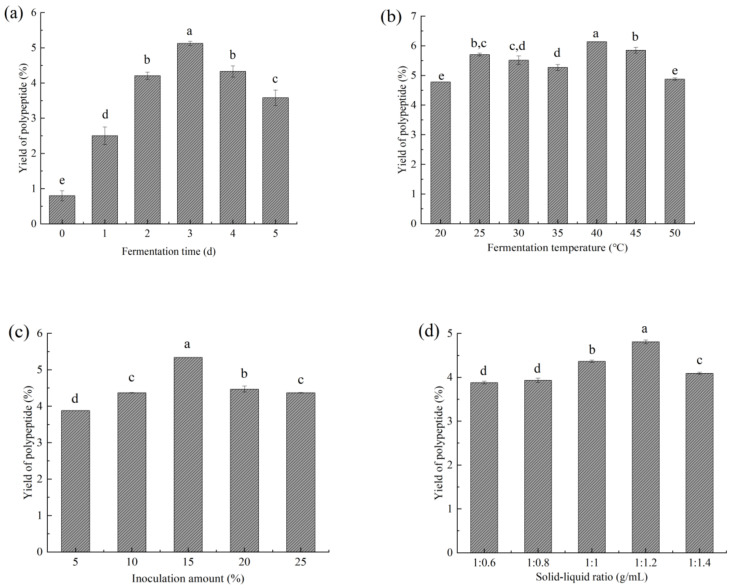
Effect of fermentation time (**a**), temperature (**b**), inoculation amount (**c**) and solid–liquid ratio (**d**) on the yield of rapeseed meal polypeptides. Different lowercase letters indicate a significant difference (*p* < 0.05).

**Figure 2 foods-11-03285-f002:**
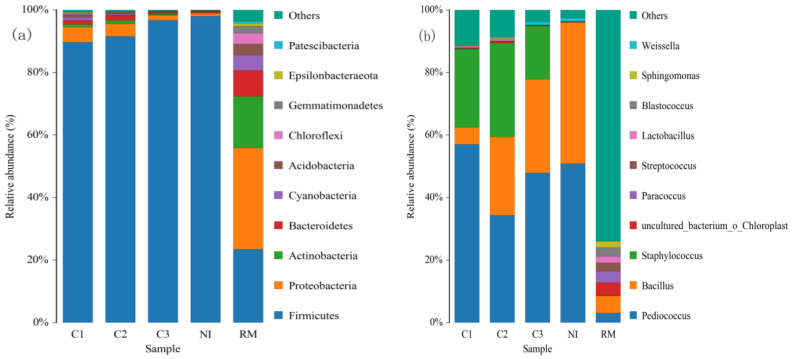
Distribution of bacterial flora of samples at phylum level (**a**) and genus level (**b**). RM is rapeseed meal, NI is rapeseed meal with water, C1, C2, C3 are fermented rapeseed meal for 24 h, 48 h, 72 h.

**Figure 3 foods-11-03285-f003:**
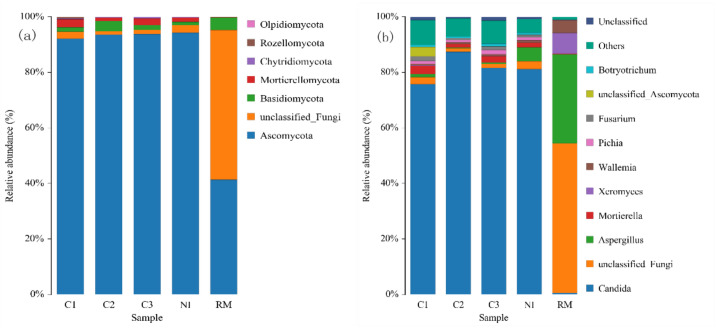
Distribution of fungi flora of samples at phylum level (**a**) and genus level (**b**).

**Table 1 foods-11-03285-t001:** Coded and actual levels of the independent variables for the design of the CCD experiment.

Independent Variables	Symbols	Coded Levels		
		−1	0	1
Fermentation temperature (°C)	A	35	40	45
Solid–liquid ratio (g/mL)	B	1:1	1:1.2	1:1.4
Inoculation volume (%)	C	10	15	20
Fermentation time (d)	D	3	4	5

**Table 2 foods-11-03285-t002:** CCD experimental design and yield of polypeptide.

Run	*A*: (°C)	*B*: (g/mL)	*C*: (%)	*D*: (d)	*PY*: (%)	*AY*: (%)
1	−1	0	1	0	6.24	6.29
2	0	0	0	0	7.39	7.43
3	0	1	1	0	5.98	5.90
4	0	1	0	−1	5.79	5.80
5	1	0	0	−1	6.19	6.10
6	−1	0	0	1	5.66	5.69
7	0	−1	−1	0	6.00	6.03
8	1	0	0	1	5.72	5.59
9	0	0	−1	−1	5.75	5.78
10	1	0	−1	0	6.19	6.17
11	−1	1	0	0	6.09	6.03
12	1	0	1	0	6.38	6.45
13	0	1	0	1	5.24	5.30
14	0	0	0	0	7.39	7.41
15	0	0	−1	1	5.28	5.29
16	0	−1	0	1	5.55	5.58
17	1	−1	0	0	6.38	6.46
19	0	0	1	−1	5.86	5.87
19	−1	0	−1	0	6.11	6.08
20	−1	0	0	−1	6.03	6.10
21	0	1	−1	0	5.72	5.70
22	−1	−1	0	0	6.28	6.21
23	0	−1	0	−1	5.84	5.81
24	0	−1	1	0	6.06	6.03
25	0	0	0	0	7.39	7.31
26	0	0	1	1	5.50	5.49
27	1	1	0	0	6.20	6.29
28	0	0	0	0	7.39	7.37
29	0	0	0	0	7.39	7.41

*A**, B**, C**, D* represent fermentation temperature, solid–liquid ratio, inoculation amount and fermentation time, *PY* represents the predicted polypeptide yield, *AY* represents the actual polypeptide yield.

**Table 3 foods-11-03285-t003:** Analysis of variance for quadratic regression equation.

Mean	*R* * ^2^ *	*Adj-R^2^*	Standard Deviations	Coefficient of Variation (%)
12.340	0.993	0.985	0.156	1.260

**Table 4 foods-11-03285-t004:** Changes in pH, titratable acidity and glucosinolate during mixed fermentation of rapeseed meal.

	0 h	24 h	48 h	72 h
pH	5.315 ± 0.01 ^a^	5.29 ± 0 ^b^	5.235 ± 0.005 ^c^	5.145 ± 0.005 ^d^
titratable acidity (g/kg)	57.295 ± 0.105 ^a^	58.35 ± 0.03 ^b^	68.875 ± 0.025 ^c^	85.54 ± 0.32 ^d^
Glucosinolate content (µmol/g)	42.33 ± 0.11 ^a^	37.605 ± 0.605 ^b^	26.925 ± 0.455 ^c^	22.775 ± 0.275 ^d^

The lowercase letters a, b, c and d represent distinctive identifies (*p* < 0.05).

**Table 5 foods-11-03285-t005:** Bacterial alpha diversity index.

Sample	OTUs	ACE	Chao1	Simpson	Shannon	Coverage
RM	496	508.53	522.25	0.99	8.04	99.97%
NI	335	374.14	374.00	0.67	2.23	99.91%
C1	343	355.99	359.87	0.61	2.55	99.97%
C2	401	412.83	425.80	0.74	2.81	99.95%
C3	305	316.29	321.11	0.66	2.09	99.96%

**Table 6 foods-11-03285-t006:** Fungal alpha diversity index.

Sample	OTUs	ACE	Chao1	Simpson	Shannon	Coverage
RM	60	63.29	63.75	0.68	2.48	99.99%
NI	98	98.00	98.00	0.34	1.64	100.00%
C1	128	128.29	128.00	0.42	2.23	100.00%
C2	96	96.00	96.00	0.24	1.19	100.00%
C3	121	121.00	121.00	0.33	1.80	100.00%

## Data Availability

The datasets generated for this study are available on request to the corresponding author.

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
