# Peer review of "Mixed-Strain Fermentation Conditions Screening of Polypeptides from Rapeseed Meal and the Microbial Diversity Analysis by High-Throughput Sequencing"

_foods, 2022, doi:10.3390/foods11203285_

Round 1

Reviewer 1 Report

Comments for authors:

Abstract:

Line 19-31: This information is not needed in the abstract.

Line 34 “the potential of the application of rapeseed meal.” should be revised to “the potential of rapeseed meal”

1. Introduction:

Line 40: Please write the reference for China ranks third with 14.05 million tons 40 in 2020/2021.

Line 41: The sentence “B. napus is pressed to extract rapeseed oil” should not start with an abbreviation.

Line 49: environmentally friendly should be replaced with environment friendly.

Line 49: Please explain clearly why microbial fermentation is superior than enzymatic hydrolysis?

Line 51:Microbial fermentation makes use of microorganisms to ferment raw materials should be corrected as Microbial fermentation uses microorganisms to ferment raw materials

 Line 51-53: Microbial fermentation makes use of microorganisms to ferment raw materials and improve the quality of fermented materials through series of physiological and biochemical reactions during the growth and development of microorganisms.

The sentence is too long. Please rephrase.

Line 74-75: it was found that the silage quality was significantly improved after fermentation and the chemical composition was also modified.

Please rephrase it into it was found that the silage quality was significantly improved after fermentation along with modifications in chemical composition.

Line 89: how microorganism function during fermentation

should be rephrased to functionality of microorganisms during fermentation

2. Materials and Methods

Line115: “purchased” should be was purchased.

Section 2.2.

Line 122: “Single factor experiment of mixed-bacteria fermentation of rapeseed meal” should be rephrased to “single factor experiment for mixed-bacteria fermentation of rapeseed meal”

Line 126: What was the reason to consider the 60 mesh screen for getting powder as

 Line 128: 2000 r/min for 10 min

What is the unit for centrifuge speed?

 Line 137: “as well as” should be there instead of “as well”

 Line 140: 2.4 “physicochemical parameters” should be “Physicochemical parameters”

 Lines 177-179 The sentence should be rephrased as:

Origin 2021 (Originlab Corporation, North- 177 ampton, MA, USA) and SPSS 26 (SPSS Inc. Chicago, IL, USA) were used for data plotting and one-way ANOVA (p < 0.05), respectively.

3 Results 

Line 190-194 The following sentence is too lengthy. Please revise it. Also provide the reference.

“This phenomenon may be linked to the fact that nutrients in the fermentation medium were sufficient in the early stage, and thus the growth and metabolism of microorganisms were vigorous, stimulating secreted protease to breakdown proteins into smaller molecular polypeptide, which increased the polypeptide content”

Line 194-198 The following sentence is too lengthy. Please revise it.

“With the prolongation of fermentation time, the nutrients in the medium could not support the growth of microorganisms and resulting in the organisms using the available peptides or small molecular proteins to synthesize their own proteins [32], causing a gradual decrease in the polypeptide concentration.”

3.1.2 Fermentation temperature

No reference was provided to support the results.

Line 210: small molecular polypeptide should be smaller polypeptides

Line 208: “This observation s possibly linked”?? what is “s”?

Line 211-215: The following sentence is too lengthy. Please revise it.

Further, the observed reduction in polypeptide content when the inoculation amount was > 15% could be attributed to too large microbial population in the medium, making it difficult for them to maintain growth and development; and as a consequence, they consume the already produced polypeptide, resulting in a decrease in the content of the polypeptide, which is similar to the result of Hou [33] on soybean meal.

Line 218: “gradually decreases beyond” should be gradually decreased beyond

This phenomenon is could be? It should be corrected.

Line 218-223: The sentence is too lengthy. Please revise it.

In Table 3 decimal places should be uniform.

3.3 physicochemical parameters: first letter should be capital.

Line 254: “increased with duration fermentation time” should be “increased with fermentation time”

Figure 2 In the title, please provide the space, for instance, 24h should be 24 h.

Line 307: “and5.52%” should be “and 5.52%”

Line 308-309: the proportion of Staphylococcus was 30.07% and the proportion of Bacillus was 25.18%

It should be rephrased as “the proportion of Staphylococcus and Bacillus was 30.07% and 25.18%, respectively.

Line 309-313: The sentence is too lengthy. Please revise it.

4. Discussion

Line 397-402 The sentence is too lengthy. Please revise it.

5. Conclusions

Line 407: Please write the full form of CCD in the conclusion as it should be independent.

Line 414: “This study can help control the fermentation process” should be “This study can help in controlling the fermentation process”

5. Conclusion section: Future work was not projected in it.

Overall comments:

  1. English language is poor in the entire manuscript.
  2. The manuscript should be concise throughout.
  3. Critical discussions of the results should be conducted.
  4. The title of the manuscript is too big. It should be revised.
  5. The references are not uniform in the list. For instance, reference 20 has a journal name abbreviated while 21 has no full journal name. Please make all the references uniform.

Author Response

Dear Reviewer,

Thank you very much for your suggestions on the manuscript. We have carefully revised this work based on these comments and suggestions.

Reviewer 2 Report

The authors measured titratable acids. Please provide details. Why they use the term "titratable acid" instead of "titratable acidity"? Which  acid they measure?

Author Response

(The authors gave the same response as above.)

Round 2

Reviewer 1 Report

The article has been revised properly.